# On Sparse Gaussian Chain Graph Models

**Calvin McCarter**
Machine Learning Department
Carnegie Mellon University
calvinm@cmu.edu

**Seyoung Kim**
Lane Center for Computational Biology
Carnegie Mellon University
sssykim@cs.cmu.edu

## Abstract

In this paper, we address the problem of learning the structure of Gaussian chain graph models in a high-dimensional space. Chain graph models are generalizations of undirected and directed graphical models that contain a mixed set of directed and undirected edges. While the problem of sparse structure learning has been studied extensively for Gaussian graphical models and more recently for conditional Gaussian graphical models (CGGMs), there has been little previous work on the structure recovery of Gaussian chain graph models. We consider linear regression models and a re-parameterization of the linear regression models using CGGMs as building blocks of chain graph models. We argue that when the goal is to recover model structures, there are many advantages of using CGGMs as chain component models over linear regression models, including convexity of the optimization problem, computational efficiency, recovery of structured sparsity, and ability to leverage the model structure for semi-supervised learning. We demonstrate our approach on simulated and genomic datasets.

## 1 Introduction

Probabilistic graphical models have been extensively studied as a powerful tool for modeling a set of conditional independencies in a probability distribution [12]. In this paper, we are concerned with a class of graphical models, called chain graph models, that has been proposed as a generalization of undirected graphical models and directed acyclic graphical models [4, 9, 14]. Chain graph models are defined over chain graphs that contain a mixed set of directed and undirected edges but no partially directed cycles.

In particular, we study the problem of learning the structure of Gaussian chain graph models in a high-dimensional setting. While the problem of learning sparse structures from high-dimensional data has been studied extensively for other related models such as Gaussian graphical models (GGMs) [8] and more recently conditional Gaussian graphical models (CGGMs) [17, 20], to our knowledge, there is little previous work that addresses this problem for Gaussian chain graph models. Even with a known chain graph structure, current methods for parameter estimation are hindered by the presence of multiple locally optimal solutions [1, 7, 21].

Since the seminal work on conditional random fields (CRFs) [13], a general recipe for constructing chain graph models [12] has been given as using CRFs as building blocks for the model. We employ this construction for Gaussian chain graph models and propose to use the recently-introduced sparse CGGMs [17, 20] as a Gaussian equivalent of general CRFs. When the goal is to learn the model structure, we show that this construction is superior to the popular alternative approach of using linear regression as component models. Some of the key advantages of our approach are due to the fact that the sparse Gaussian chain graph models inherit the desirable properties of sparse CGGM such as convexity of the optimization problem and structured output prediction. In fact, our work is the first to introduce a joint estimation procedure for both the graph structure and parameters as a convex optimization problem, given the groups of variables for chain components. Another advan-

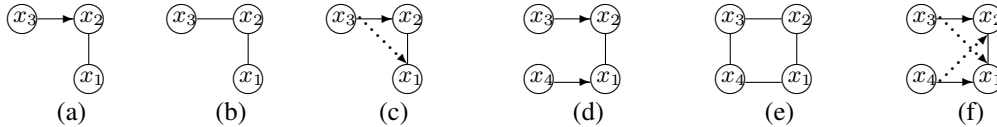

Figure 1: Illustration of chain graph models. (a) A chain graph with two components, $\{x_1, x_2\}$ and $\{x_3\}$. (b) The moralized graph of the chain graph in (a). (c) After inference in the chain graph in (a), inferred indirect dependencies are shown as the dotted line. (d) A chain graph with three components, $\{x_1, x_2\}$, $\{x_3\}$, and $\{x_4\}$. (e) The moralized graph of the chain graph in (d). (f) After inference in the chain graph in (d), inferred indirect dependencies are shown as the dotted lines.

tage of our approach is the ability to model a functional mapping from multiple related variables to other multiple related variables in a more natural way via moralization in chain graphs than other approaches that rely on complex penalty functions for inducing structured sparsity [11, 15].

Our work on sparse Gaussian chain graphs is motivated by problems in integrative genomic data analyses [6, 18]. While sparse GGMs have been extremely popular for learning networks from datasets of single modality such as gene-expression levels [8], we propose that sparse Gaussian chain graph models with CGGM components can be used to learn a cascade of networks by integrating multiple types of genomic data in a single statistical analysis. We show that our approach can reveal the module structures as well as the functional mapping between modules in different types of genomic data effectively. Furthermore, as the cost of collecting each data type differs, we show that semi-supervised learning can be used to make effective use of both fully-observed and partially-observed data.

## 2 Sparse Gaussian Chain Graph Models

We consider a chain graph model for a probability distribution over $J$ random variables $\mathbf{x} = \{x_1, \ldots, x_J\}$. The chain graph model assumes that the random variables are partitioned into $C$ chain components $\{\mathbf{x}_1, \ldots, \mathbf{x}_C\}$, the $\tau$th component having size $|\tau|$. In addition, it assumes a partially directed graph structure, where edges between variables within each chain component are undirected and edges across two chain components are directed. Given this chain graph structure, the joint probability distribution factorizes as follows:

$$p(\mathbf{x}) = \prod_{\tau=1}^{C} p(\mathbf{x}_\tau | \mathbf{x}_{\mathrm{pa}(\tau)}),$$

where $\mathbf{x}_{\mathrm{pa}(\tau)}$ is the set of variables that are parents of one or more variables in $\mathbf{x}_\tau$. Each factor $p(\mathbf{x}_\tau | \mathbf{x}_{\mathrm{pa}(\tau)})$ models the conditional distribution of the chain component variables $\mathbf{x}_\tau$ given $\mathbf{x}_{\mathrm{pa}(\tau)}$. This model can also be viewed as being constructed with CRFs for $p(\mathbf{x}_\tau | \mathbf{x}_{\mathrm{pa}(\tau)})$'s [13].

The conditional independence properties of undirected and directed graphical models have been extended to chain graph models [9, 14]. This can be easily seen by first constructing a moralized graph, where undirected edges are added between any pairs of nodes in $\mathbf{x}_{\mathrm{pa}(\tau)}$ for each chain component $\tau$ and all the directed edges are converted into undirected edges (Figure 1). Then, subsets of variables $\mathbf{x}_a$ and $\mathbf{x}_b$ are conditionally independent given $\mathbf{x}_c$, if $\mathbf{x}_a$ and $\mathbf{x}_b$ are separated by $\mathbf{x}_c$ in the moralized graph. This conditional independence criterion for a chain graph is called $c$-separation and generalizes $d$-separation for Bayesian networks [12].

In this paper, we focus on Gaussian chain graph models, where both $p(\mathbf{x})$ and $p(\mathbf{x}_\tau | \mathbf{x}_{\mathrm{pa}(\tau)})$'s are Gaussian distributed. Below, we review linear regression models and CGGMs as chain component models, and introduce our approach for learning chain graph model structures.

### 2.1 Sparse Linear Regression as Chain Component Model

As the specific functional form of $p(\mathbf{x}_\tau | \mathbf{x}_{\mathrm{pa}(\tau)})$ in Gaussian chain graphs models, a linear regression model with multivariate responses has been widely considered [2, 3, 7]:

$$p(\mathbf{x}_\tau | \mathbf{x}_{\mathrm{pa}(\tau)}) = N(\mathbf{B}_\tau \mathbf{x}_{\mathrm{pa}(\tau)}, \boldsymbol{\Theta}_\tau^{-1}), \tag{1}$$

where $\mathbf{B}_\tau \in \mathbb{R}^{|\tau| \times |\mathrm{pa}(\tau)|}$ is the matrix of regression coefficients and $\boldsymbol{\Theta}_\tau$ is the $|\tau| \times |\tau|$ inverse covariance matrix that models correlated noise. Then, the non-zero elements in $\mathbf{B}_\tau$ indicate the

presence of directed edges from $\mathbf{x}_{\mathrm{pa}(\tau)}$ to $\mathbf{x}_\tau$, and the non-zero elements in $\mathbf{\Theta}_\tau$ correspond to the undirected edges among the variables in $\mathbf{x}_\tau$. When the graph structure is known, an iterative procedure has been proposed to estimate the model parameters, but it converges only to one of many locally-optimal solutions [7].

When the chain component model has the form of Eq. (1), in order to jointly estimate the sparse graph structure and the parameters, we adopt sparse multivariate regression with covariance estimation (MRCE) [16] for each chain component and solve the following optimization problem:

$$\min \sum_{\tau=1}^{C} \mathrm{tr}((\mathbf{X}_\tau - \mathbf{X}_{\mathrm{pa}(\tau)}\mathbf{B}_\tau^T)\mathbf{\Theta}_\tau(\mathbf{X}_\tau - \mathbf{X}_{\mathrm{pa}(\tau)}\mathbf{B}_\tau^T)^T) - N \log |\mathbf{\Theta}_\tau| + \lambda \sum_{\tau=1}^{C} ||\mathbf{B}_\tau||_1 + \gamma \sum_{\tau=1}^{C} ||\mathbf{\Theta}_\tau||_1,$$

where $\mathbf{X}_\alpha \in \mathbb{R}^{N \times |\alpha|}$ is a dataset for $N$ samples, $|| \cdot ||_1$ is the sparsity-inducing $L_1$ penalty, and $\lambda$ and $\gamma$ are the regularization parameters that control the amount of sparsity in the parameters. As in MRCE [16], the problem above is not convex, but only bi-convex.

## 2.2 Sparse Conditional Gaussian Graphical Model as Chain Component Model

As an alternative model for $p(\mathbf{x}_\tau|\mathbf{x}_{\mathrm{pa}(\tau)})$ in Gaussian chain graph models, a re-parameterization of the linear regression model in Eq. (1) with natural parameters has been considered [14]. This model also has been called a CGGM [17] or Gaussian CRF [20] due to its equivalence to a CRF. A CGGM for $p(\mathbf{x}_\tau|\mathbf{x}_{\mathrm{pa}(\tau)})$ takes the standard form of undirected graphical models as a log-linear model:

$$p(\mathbf{x}_\tau|\mathbf{x}_{\mathrm{pa}(\tau)}) = \exp\left(-\frac{1}{2}\mathbf{x}_\tau^T\mathbf{\Theta}_\tau\mathbf{x}_\tau - \mathbf{x}_\tau^T\mathbf{\Theta}_{\tau,\mathrm{pa}(\tau)}\mathbf{x}_{\mathrm{pa}(\tau)}\right)/A(\mathbf{x}_{\mathrm{pa}(\tau)}), \qquad (2)$$

where $\mathbf{\Theta}_\tau \in \mathbb{R}^{|\tau| \times |\tau|}$ and $\mathbf{\Theta}_{\tau,\mathrm{pa}(\tau)} \in \mathbf{R}^{|\tau| \times |\mathrm{pa}(\tau)|}$ are the parameters for the feature weights between pairs of variables within $\mathbf{x}_\tau$ and between pairs of variables across $\mathbf{x}_\tau$ and $\mathbf{x}_{\mathrm{pa}(\tau)}$, respectively, and $A(\mathbf{x}_{\mathrm{pa}(\tau)})$ is the normalization constant. The non-zero elements of $\mathbf{\Theta}_\tau$ and $\mathbf{\Theta}_{\tau,\mathrm{pa}(\tau)}$ indicate edges among the variables in $\mathbf{x}_\tau$ and between $\mathbf{x}_\tau$ and $\mathbf{x}_{\mathrm{pa}(\tau)}$, respectively.

The linear regression model in Eq. (1) can be viewed as the result of performing inference in the probabilistic graphical model given by the CGGM in Eq. (2). This relationship between the two models can be seen by re-writing Eq. (2) in the form of a Gaussian distribution:

$$p(\mathbf{x}_\tau|\mathbf{x}_{\mathrm{pa}(\tau)}) = N(-\mathbf{\Theta}_\tau^{-1}\mathbf{\Theta}_{\tau,\mathrm{pa}(\tau)}\mathbf{x}_{\mathrm{pa}(\tau)}, \mathbf{\Theta}_\tau^{-1}), \qquad (3)$$

where marginalization in a CGGM involves computing $\mathbf{B}_\tau\mathbf{x}_{\mathrm{pa}(\tau)} = -\mathbf{\Theta}_\tau^{-1}\mathbf{\Theta}_{\tau,\mathrm{pa}(\tau)}\mathbf{x}_{\mathrm{pa}(\tau)}$ to obtain a linear regression model parameterized by $\mathbf{B}_\tau$.

In order to estimate the graph structure and parameters for Gaussian chain graph models with CGGMs as chain component models, we adopt the procedure for learning a sparse CGGM [17, 20] and minimize the negative log-likelihood of data along with sparsity-inducing $L_1$ penalty:

$$\min -\mathcal{L}(\mathbf{X}; \mathbf{\Theta}) + \lambda \sum_{\tau=1}^{C} ||\mathbf{\Theta}_{\tau,\mathrm{pa}(\tau)}||_1 + \gamma \sum_{\tau=1}^{C} ||\mathbf{\Theta}_\tau||_1,$$

where $\mathbf{\Theta} = \{\mathbf{\Theta}_\tau, \mathbf{\Theta}_{\tau,\mathrm{pa}(\tau)}, \tau = 1, \ldots, C\}$ and $\mathcal{L}(\mathbf{X}; \mathbf{\Theta})$ is the data log-likelihood for dataset $\mathbf{X} \in \mathbb{R}^{N \times J}$ for $N$ samples. Unlike MRCE, the optimization problem for a sparse CGGM is convex, and efficient algorithms have been developed to find the globally-optimal solution with substantially lower computation time than that for MRCE [17, 20].

While maximum likelihood estimation leads to the equivalent parameter estimates for CGGMs and linear regression models via the transformation $\mathbf{B}_\tau = -\mathbf{\Theta}_\tau^{-1}\mathbf{\Theta}_{\tau,\mathrm{pa}(\tau)}$, imposing a sparsity constraint on each model leads to different estimates for the sparsity pattern of the parameters and the model structure [17]. The graph structure of a sparse CGGM directly encodes the probabilistic dependencies among the variables, whereas the sparsity pattern of $\mathbf{B}_\tau = -\mathbf{\Theta}_\tau^{-1}\mathbf{\Theta}_{\tau,\mathrm{pa}(\tau)}$ obtained after marginalization can be interpreted as indirect influence of covariates $\mathbf{x}_{\mathrm{pa}(\tau)}$ on responses $\mathbf{x}_\tau$. As illustrated in Figures 1(c) and 1(f), the CGGM parameters $\mathbf{\Theta}_{\tau,\mathrm{pa}(\tau)}$ (directed edges with solid line) can be interpreted as direct dependencies between pairs of variables across $\mathbf{x}_\tau$ and $\mathbf{x}_{\mathrm{pa}(\tau)}$, whereas $\mathbf{B}_\tau = -\mathbf{\Theta}_\tau^{-1}\mathbf{\Theta}_{\tau,\mathrm{pa}(\tau)}$ obtained from inference can be viewed as indirect and inferred dependencies (directed edges with dotted line).

We argue in this paper that when the goal is to learn the model structure, performing the estimation with CGGMs for chain component models can lead to a more meaningful representation of the underlying structure in data than imposing a sparsity constraint on linear regresssion models. Then the corresponding linear regression model can be inferred via marginalization. This approach also inherits many of the advantages of sparse CGGMs such as convexity of optimization problem.

### 2.3 Markov Properties and Chain Component Models

When a CGGM is used as the component model, the overall chain graph model is known to have Lauritzen-Wermuth-Frydenberg (LWF) Markov properties [9]. The LWF Markov properties also correspond to the standard probabilistic independencies in more general chain graphs constructed by using CRFs as building blocks [12].

Many previous works have noted that LWF Markov properties do not hold for the chain graph models with linear regression models [2, 3]. The alternative Markov properties (AMP) were therefore introduced as the set of probabilistic independencies associated with chain graph models with linear regression component models [2, 3]. It has been shown that the LWF and AMP Markov properties are equivalent only for chain graph structures that do not contain the graph in Figure 1(a) as a subgraph [2, 3]. For example, according to the LWF Markov property, in the chain graph model in Figure 1(a), $x_1 \perp x_3 | x_2$ as $x_1$ and $x_3$ are separated by $x_2$ in the moralized graph in Figure 1(b). However, the corresponding AMP Markov property implies a different probabilistic independence relationship, $x_1 \perp x_3$. In the model in Figure 1(d), according to the LWF Markov property, we have $x_1 \perp x_3 | \{x_2, x_4\}$, whereas the AMP Markov property gives $x_1 \perp x_3 | x_4$.

We observe that when using sparse CGGMs as chain component models, we estimate a model with the LWF Markov properties and perform marginalization in this model to obtain a model with linear-regression chain components that can be interpreted with the AMP Markov properties.

## 3 Sparse Two-Layer Gaussian Chain Graph Models for Structured Sparsity

Another advantage of using CGGMs as chain component models instead of linear regression is that the moralized graph, which is used to define the LWF Markov properties, can be leveraged to discover the underlying structure in a correlated functional mapping from multiple inputs to multiple outputs. In this section, we show that a sparse two-layer Gaussian chain graph model with CGGM components can be used to learn structured sparsity. The key idea behind our approach is that while inference in CGGMs within the chain graph model can reveal the shared sparsity patterns for multiple related outputs, a moralization of the chain graph can reveal those for multiple inputs.

Statistical methods for learning models with structured sparsity were extensively studied in the literature of multi-task learning, where the goal is to find input features that influence multiple related outputs simultaneously [5, 11, 15]. Most of the previous works assumed the output structure to be known *a priori*. Then, they constructed complex penalty functions that leverage this known output structure, in order to induce structured sparsity pattern in the estimated parameters in linear regression models. In contrast, a sparse CGGM was proposed as an approach for performing a joint estimation of the output structure and structured sparsity for multi-task learning. As was discussed in Section 2.2, once the CGGM structure is estimated, the inputs relevant for multiple related outputs could be revealed via probabilistic inference in the graphical model.

While sparse CGGMs focused on leveraging the output structure for improved predictions, another aspect of learning structured sparsity is to consider the input structure to discover multiple related inputs jointly influencing an output. As CGGM is a discriminative model that does not model the input distribution, it is unable to capture input relatedness directly, although discriminative models in general are known to improve prediction accuracy. We address this limitation of CGGMs by embedding CGGMs within a chain graph and examining the moralized graph.

We set up a two-layer Gaussian chain graph model for inputs $\mathbf{x}$ and outputs $\mathbf{y}$ as follows:

$$p(\mathbf{y}, \mathbf{x}) = p(\mathbf{y}|\mathbf{x})p(\mathbf{x}) = \left( \exp(-\frac{1}{2}\mathbf{y}^T \mathbf{\Theta_{yy}} \mathbf{y} - \mathbf{x}^T \mathbf{\Theta_{xy}} \mathbf{y})/A_1(\mathbf{x}) \right) \left( \exp(-\frac{1}{2}\mathbf{x}^T \mathbf{\Theta_{xx}} \mathbf{x})/A_2 \right),$$

where a CGGM is used for $p(\mathbf{y}|\mathbf{x})$ and a GGM for $p(\mathbf{x})$, and $A_1(\mathbf{x})$ and $A_2$ are normalization constants. As the full model factorizes into two factors $p(\mathbf{y}|\mathbf{x})$ and $p(\mathbf{x})$ with distinct sets of parameters,

a sparse graph structure and parameters can be learned by using the optimization methods for sparse CGGM [20] and sparse GGM [8, 10].

The estimated Gaussian chain graph model leads to a GGM over both the inputs and outputs, which reveals the structure of the moralized graph:

$$p(\mathbf{y}, \mathbf{x}) = N\left(\mathbf{0}, \begin{pmatrix} \mathbf{\Theta_{yy}} & \mathbf{\Theta_{xy}^T} \\ \mathbf{\Theta_{xy}} & \mathbf{\Theta_{xx}} + \mathbf{\Theta_{xy}\Theta_{yy}^{-1}\Theta_{xy}^T} \end{pmatrix}^{-1}\right).$$

In the above GGM, we notice that the graph structure over inputs $\mathbf{x}$ consists of two components, one for $\mathbf{\Theta_{xx}}$ describing the conditional dependencies within the input variables and another for $\mathbf{\Theta_{xy}\Theta_{yy}^{-1}\Theta_{xy}^T}$ that reflects the results of moralization in the chain graph. If the graph $\mathbf{\Theta_{yy}}$ contains connected components, the operation $\mathbf{\Theta_{xy}\Theta_{yy}^{-1}\Theta_{xy}^T}$ for moralization induces edges among those inputs influencing the outputs in each connected component.

Our approach is illustrated in Figure 2. Given the model in Figure 2(a), Figure 2(b) illustrates the inferred structured sparsity for a functional mapping from multiple inputs to multiple outputs. In Figure 2(b), the dotted edges correspond to inferred indirect dependencies introduced via marginalization in the CGGM $p(\mathbf{y}|\mathbf{x})$, which reveals how each input is influencing multiple related outputs. On the other hand, the additional edges among $x_j$'s have been introduced by

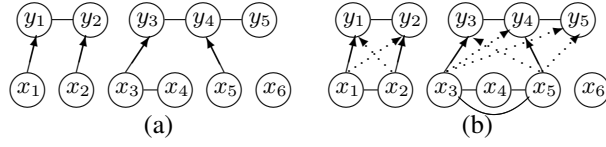

(a)                    (b)

Figure 2: Illustration of sparse two-layer Gaussian chain graphs with CGGMs. (a) A two-layer Gaussian chain graph. (b) The results of performing inference and moralization in (a). The dotted edges correspond to indirect dependencies inferred by inference. The edges among $x_j$'s represent the dependencies introduced by moralization.

moralization $\mathbf{\Theta_{xy}\Theta_{yy}^{-1}\Theta_{xy}^T}$ for multiple inputs jointly influencing each output. Combining the results of marginalization and moralization, the two connected components in Figure 2(b) represent the functional mapping from $\{x_1, x_2\}$ to $\{y_1, y_2\}$ and from $\{x_3, x_4, x_5\}$ to $\{y_3, y_4, y_5\}$, respectively.

## 4 Sparse Multi-layer Gaussian Chain Graph Models

In this section, we extend the two-layer Gaussian chain graph model from the previous section into a multi-layer model to model data that are naturally organized into multiple layers. Our approach is motivated by problems in integrative genomic data analysis. In order to study the genetic architecture of complex diseases, data are often collected for multiple data types, such as genotypes, gene expressions, and phenotypes for a population of individuals [6, 18]. The primary goal of such studies is to identify the genotype features that influence gene expressions, which in turn influence phenotypes. In such problems, data can be naturally organized into multiple layers, where the influence of features in each layer propagates to the next layer in sequence. In addition, it is well-known that the expressions of genes within the same functional module are correlated and influenced by the common genotype features and that the coordinated expressions of gene modules affect multiple related phenotypes jointly. These underlying structures in the genomic data can be potentially revealed by inference and moralization in sparse Gaussian chain graph models with CGGM components.

In addition, we explore the use of semi-supervised learning, where the top and bottom layer data are fully observed but the middle-layer data are collected only for a subset of samples. In our application, genotype data and phenotype data are relatively easy to collect from patients' blood samples and from observations. However, gene-expression data collection is more challenging, as invasive procedure such as surgery or biopsy is required to obtain tissue samples.

### 4.1 Models

Given variables, $\mathbf{x} = \{x_1, \ldots, x_J\}$, $\mathbf{y} = \{y_1, \ldots, y_K\}$, and $\mathbf{z} = \{z_1, \ldots, z_L\}$, at each of the three layers, we set up a three-layer Gaussian chain graph model as follows:

$$p(\mathbf{z}, \mathbf{y}|\mathbf{x}) = p(\mathbf{z}|\mathbf{y})p(\mathbf{y}|\mathbf{x})$$

$$= \left(\exp(-\frac{1}{2}\mathbf{z}^T\mathbf{\Theta_{zz}z} - \mathbf{y}^T\mathbf{\Theta_{yz}z})/C_2(\mathbf{y})\right)\left(\exp(-\frac{1}{2}\mathbf{y}^T\mathbf{\Theta_{yy}y} - \mathbf{x}^T\mathbf{\Theta_{xy}y})/C_1(\mathbf{x})\right), \quad (4)$$

where $C_1(\mathbf{x})$ and $C_2(\mathbf{y})$ are the normalization constants. In our application, $\mathbf{x}$, $\mathbf{y}$, and $\mathbf{z}$ correspond to genotypes, gene-expression levels, and phenotypes, respectively. As the focus of such studies lies on discovering how the genotypic variability influences gene expressions and phenotypes rather than the structure in genotype features, we do not model $p(\mathbf{x})$ directly.

Given the estimated sparse model for Eq. (4), structured sparsity pattern can be recovered via inference and moralization. Computing $\mathbf{B_{xy}} = -\boldsymbol{\Theta}_{\mathbf{yy}}^{-1}\boldsymbol{\Theta}_{\mathbf{xy}}^T$ and $\mathbf{B_{yz}} = -\boldsymbol{\Theta}_{\mathbf{zz}}^{-1}\boldsymbol{\Theta}_{\mathbf{yz}}^T$ corresponds to performing inference to reveal how multiple related $y_k$'s in $\boldsymbol{\Theta}_{\mathbf{yy}}$ (or $z_l$'s in $\boldsymbol{\Theta}_{\mathbf{zz}}$) are jointly influenced by a common set of relevant $x_j$'s (or $y_k$'s). On the other hand, the effects of moralization can be seen from the joint distribution $p(\mathbf{z}, \mathbf{y}|\mathbf{x})$ derived from Eq. (4):

$$p(\mathbf{z}, \mathbf{y}|\mathbf{x}) = N(-\boldsymbol{\Theta}_{(\mathbf{zz},\mathbf{yy})}^{-1}\boldsymbol{\Theta}_{(\mathbf{yz},\mathbf{xy})}^T\mathbf{x}, \boldsymbol{\Theta}_{(\mathbf{zz},\mathbf{yy})}^{-1}),$$

where $\boldsymbol{\Theta}_{(\mathbf{yz},\mathbf{xy})} = (\mathbf{0}_{J\times L}, \boldsymbol{\Theta}_{\mathbf{xy}})$ and $\boldsymbol{\Theta}_{(\mathbf{zz},\mathbf{yy})} = \begin{pmatrix} \boldsymbol{\Theta}_{\mathbf{zz}} & \boldsymbol{\Theta}_{\mathbf{yz}}^T \\ \boldsymbol{\Theta}_{\mathbf{yz}} & \boldsymbol{\Theta}_{\mathbf{yy}} + \boldsymbol{\Theta}_{\mathbf{yz}}\boldsymbol{\Theta}_{\mathbf{zz}}^{-1}\boldsymbol{\Theta}_{\mathbf{yz}}^T \end{pmatrix}$. $\boldsymbol{\Theta}_{(\mathbf{zz},\mathbf{yy})}$ corresponds to the undirected graphical model over $\mathbf{z}$ and $\mathbf{y}$ conditional on $\mathbf{x}$ after moralization.

## 4.2 Semi-supervised Learning

Given a dataset $\mathcal{D} = \{\mathcal{D}_o, \mathcal{D}_h\}$, where $\mathcal{D}_o = \{\mathbf{X}_o, \mathbf{Y}_o, \mathbf{Z}_o\}$ for the fully-observed data and $\mathcal{D}_h = \{\mathbf{X}_h, \mathbf{Z}_h\}$ for the samples with missing gene-expression levels, for semi-supervised learning, we adopt an EM algorithm that iteratively maximizes the expected log-likelihood of complete data:

$$\mathcal{L}(\mathcal{D}_o; \boldsymbol{\Theta}) + \mathrm{E}\big[\mathcal{L}(\mathcal{D}_h, \mathbf{Y}_h; \boldsymbol{\Theta})\big],$$

combined with $L_1$-regularization, where $\mathcal{L}(\mathcal{D}_o; \boldsymbol{\Theta})$ is the data log-likelihood with respect to the model in Eq. (4) and the expectation is taken with respect to:

$$p(\mathbf{y}|\mathbf{z}, \mathbf{x}) = N(\mu_{\mathbf{y}|\mathbf{x},\mathbf{z}}, \boldsymbol{\Sigma}_{\mathbf{y}|\mathbf{x},\mathbf{z}}),$$

$$\mu_{\mathbf{y}|\mathbf{x},\mathbf{z}} = -\boldsymbol{\Sigma}_{\mathbf{y}|\mathbf{x},\mathbf{z}}(\boldsymbol{\Theta}_{\mathbf{yz}}\mathbf{z} + \boldsymbol{\Theta}_{\mathbf{xy}}^T\mathbf{x}) \quad \text{and} \quad \boldsymbol{\Sigma}_{\mathbf{y}|\mathbf{x},\mathbf{z}} = (\boldsymbol{\Theta}_{\mathbf{yy}} + \boldsymbol{\Theta}_{\mathbf{yz}}\boldsymbol{\Theta}_{\mathbf{zz}}^{-1}\boldsymbol{\Theta}_{\mathbf{yz}}^T)^{-1}.$$

# 5 Results

In this section, we empirically demonstrate that CGGMs are more effective components for sparse Gaussian chain graph models than linear regression for various tasks, using synthetic and real-world genomic datasets. We used the sparse three-layer structure for $p(\mathbf{z}, \mathbf{y}|\mathbf{x})$ in all our experiments.

## 5.1 Simulation Study

In simulation study, we considered two scenarios for true models, CGGM-based and linear-regression-based Gaussian chain graph models. We evaluated the performance in terms of graph structure recovery and prediction accuracy in both supervised and semi-supervised settings.

In order to simulate data, we assumed the problem size of $J$=500, $K$=100, and $L$=50 for $\mathbf{x}$, $\mathbf{y}$, and $\mathbf{z}$, respectively, and generated samples from known true models. Since we do not model $p(\mathbf{x})$, we used an arbitrary choice of multinomial distribution to generate samples for $\mathbf{x}$. The true parameters for CGGM-based simulation were set as follows. We set the graph structure in $\boldsymbol{\Theta}_{\mathbf{yy}}$ to a randomly-generated scale-free network with a community structure [19] with six communities. The edge weights were drawn randomly from a uniform distribution [0.8, 1.2]. We then set $\boldsymbol{\Theta}_{\mathbf{yy}}$ to the graph Laplacian of this network plus small positive values along the diagonal so that $\boldsymbol{\Theta}_{\mathbf{yy}}$ is positive definite. We generated $\boldsymbol{\Theta}_{\mathbf{zz}}$ using a similar strategy, assuming four communities. $\boldsymbol{\Theta}_{\mathbf{xy}}$ was set to a sparse random matrix, where 0.4% of the elements have non-zero values drawn from a uniform distribution [-1.2,-0.8]. $\boldsymbol{\Theta}_{\mathbf{yz}}$ was generated using a similar strategy, with a sparsity level of 0.5%. We set the sparsity pattern of $\boldsymbol{\Theta}_{\mathbf{yz}}$ so that it roughly respects the functional mapping from communities in $\mathbf{y}$ to communities in $\mathbf{z}$. Specifically, after reordering the variables in $\mathbf{y}$ and $\mathbf{z}$ by performing hierarchical clustering on each of the two networks $\boldsymbol{\Theta}_{\mathbf{yy}}$ and $\boldsymbol{\Theta}_{\mathbf{zz}}$, the non-zero elements were selected randomly around the diagonal of $\boldsymbol{\Theta}_{\mathbf{yz}}$.

We set the true parameters for the linear-regression-based models using the same strategy as the CGGM-based simulation above for $\boldsymbol{\Theta}_{\mathbf{yy}}$ and $\boldsymbol{\Theta}_{\mathbf{zz}}$. We set $\mathbf{B_{xy}}$ so that 50% of the variables in $\mathbf{x}$ have non-zero influence on five randomly chosen variables in $\mathbf{y}$ in one randomly chosen community in $\boldsymbol{\Theta}_{\mathbf{yy}}$. We set $\mathbf{B_{yz}}$ in a similar manner, assuming 80% of the variables in $\mathbf{y}$ are relevant to eight randomly-chosen variables in $\mathbf{z}$ from a randomly-chosen community in $\boldsymbol{\Theta}_{\mathbf{zz}}$.

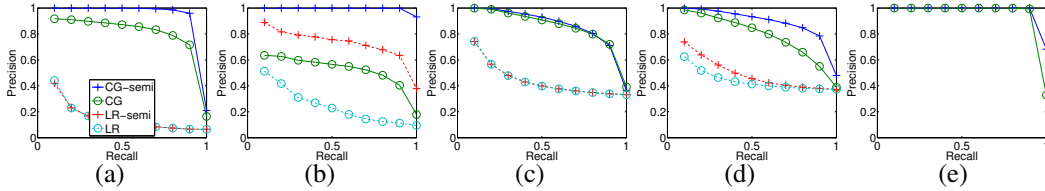

(a)       (b)       (c)       (d)       (e)

Figure 4: Precision/recall curves for graph structure recovery in CGGM-based simulation study. (a) $\Theta_{yy}$, (b) $\Theta_{zz}$, (c) $B_{xy}$, (d) $B_{yz}$, and (e) $\Theta_{xy}$. (CG: CGGM-based models with supervised learning, CG-semi: CG with semi-supervised learning, LR: linear-regression-based models with supervised learning, LR-semi: LR with semi-supervised learning.)

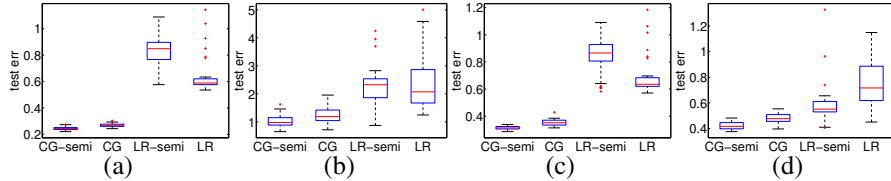

(a)       (b)       (c)       (d)

Figure 5: Prediction errors in CGGM-based simulation study. The same estimated models in Figure 4 were used to predict (a) $y$ given $x, z$, (b) $z$ given $x$, (c) $y$ given $x$, and (d) $z$ given $y$.

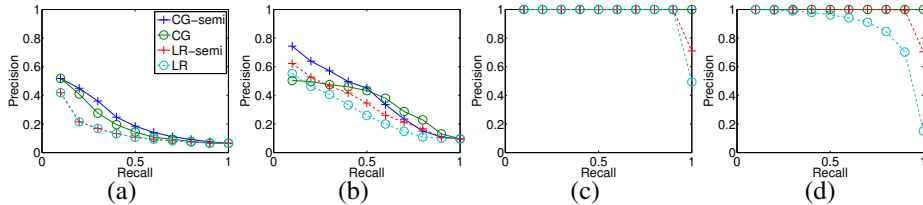

(a)       (b)       (c)       (d)

Figure 6: Performance for graph structure recovery in linear-regression-based simulation study. Precision/recall curves are shown for (a) $\Theta_{yy}$, (b) $\Theta_{zz}$, (c) $B_{xy}$, and (d) $B_{yz}$.

Each dataset consisted of 600 samples, of which 400 and 200 samples were used as training and test sets. To select the regularization parameters, we estimated a model using 300 samples, evaluated prediction errors on the other 100 samples in the training set, and selected the values with the lowest prediction errors. We used the optimization methods in [20] for CGGM-based models and the MRCE procedure [16] for linear-regression-based models.

Figure 3 illustrates how the model with CGGM chain components can be used to discover the structured sparsity via inference and moralization. In each panel, black and bright pixels correspond to zero and non-zero values, respectively. While Figure 3(a) shows how variables in $z$ are related in $\Theta_{zz}$, Figure 3(b) shows $B_{yz} = -\Theta_{zz}^{-1}\Theta_{yz}^{T}$ obtained via marginalization within the CGGM $p(z|y)$, where functional mappings from variables in $y$ to multiple related variables

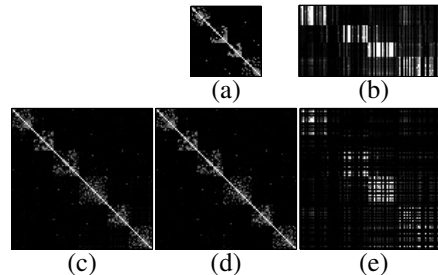

(a)       (b)

(c)       (d)       (e)

Figure 3: Illustration of the structured sparsity recovered by the model with CGGM components, simulated dataset. (a) $\Theta_{zz}$. (b) $B_{yz} = -\Theta_{zz}^{-1}\Theta_{yz}^{T}$ shows the effects of marginalization (white vertical bars). The effects of moralization are shown in (c) $\Theta_{yy} + \Theta_{yz}\Theta_{zz}^{-1}\Theta_{yz}^{T}$, and its decomposition into (d) $\Theta_{yy}$ and (e) $\Theta_{yz}\Theta_{zz}^{-1}\Theta_{yz}^{T}$.

in $z$ can be seen as white vertical bars. In Figure 3(c), the effects of moralization $\Theta_{yy} + \Theta_{yz}\Theta_{zz}^{-1}\Theta_{yz}^{T}$ are shown, which further decomposes into $\Theta_{yy}$ (Figure 3(d)) and $\Theta_{yz}\Theta_{zz}^{-1}\Theta_{yz}^{T}$ (Figure 3(e)). The additional edges among variables in $y$ in Figure 3(e) correspond to the edges introduced via moralization and show the groupings of the variables $y$ as the block structure along the diagonal. By examining Figures 3(b) and 3(e), we can infer a functional mapping from modules in $y$ to modules in $z$.

In order to systematically compare the performance of the two types of models, we examined the average performance over 30 randomly-generated datasets. We considered both supervised and semi-supervised settings. Assuming that 200 samples out of the total 400 training samples were

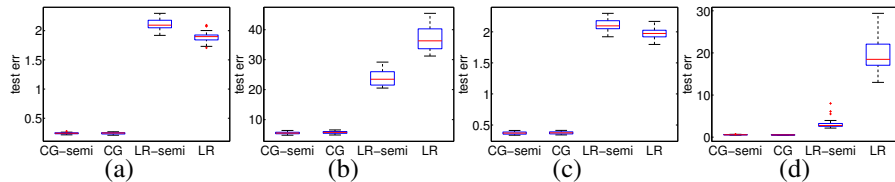

Figure 7: Prediction errors in linear-regression-based simulation study. The same estimated models in Figure 6 were used to predict (a) $\mathbf{y}$ given $\mathbf{x}, \mathbf{z}$, (b) $\mathbf{z}$ given $\mathbf{x}$, (c) $\mathbf{y}$ given $\mathbf{x}$, and (d) $\mathbf{z}$ given $\mathbf{y}$.

missing data for $\mathbf{y}$, for supervised learning, we used only those samples with complete data; for semi-supervised learning, we used all samples, including partially-observed cases.

The precision/recall curves for recovering the true graph structures are shown in Figure 4, using datasets simulated from the true models with CGGM components. Each curve was obtained as an average over 30 different datasets. We observe that in both supervised and semi-supervised settings, the models with CGGM components outperform the ones with linear regression components. In addition, the performance of the CGGM-based models improves significantly, when using the partially-observed data in addition to the fully-observed samples (the curve for CG-semi in Figure 4), compared to using only the fully-observed samples (the curve for CG in Figure 4). This improvement from using partially-observed data is substantially smaller for the linear-regression-based models. The average prediction errors from the same set of estimated models in Figure 4 are shown in Figure 5. The CGGM-based models outperform in all prediction tasks, because they can leverage the underlying structure in the data and estimate models more effectively.

For the simulation scenario using the linear-regression-based true models, we show the results for precision/recall curves and prediction errors in Figures 6 and 7, respectively. We find that even though the data were generated from chain graph models with linear regression components, the CGGM-based methods perform as well as or better than the other models.

## 5.2 Integrative Genomic Data Analysis

We applied the two types of three-layer chain graph models to single-nucleotide-polymorphism (SNP), gene-expression, and phenotype data from the pancreatic islets study for diabetic mice [18]. We selected 200 islet gene-expression traits after performing hierarchical clustering to find several gene modules. Our dataset also included 1000 SNPs and 100 pancreatic islet cell phenotypes. Of the total 506 samples, we

Table 1: Prediction errors, mouse diabetes data

| Task | CG-semi | CG | LR-semi | LR |
|---|---|---|---|---|
| $\mathbf{y} \mid \mathbf{x}, \mathbf{z}$ | 0.9070 | 0.9996 | 1.0958 | 0.9671 |
| $\mathbf{z} \mid \mathbf{x}$ | 1.0661 | 1.0585 | 1.0505 | 1.0614 |
| $\mathbf{y} \mid \mathbf{x}$ | 0.8989 | 0.9382 | 0.9332 | 0.9103 |
| $\mathbf{z} \mid \mathbf{y}$ | 1.0712 | 1.0861 | 1.1095 | 1.0765 |

used 406 as training set, of which 100 were held out as a validation set to select regularization parameters, and used the remaining 100 samples as test set to evaluate prediction accuracies. We considered both supervised and semi-supervised settings, assuming gene expressions are missing for 150 mice. In supervised learning, only those samples without missing gene expressions were used.

As can be seen from the prediction errors in Table 1, the models with CGGM chain components are more accurate in various prediction tasks. In addition, the CGGM-based models can more effectively leverage the samples with partially-observed data than linear-regression-based models.

# 6 Conclusions

In this paper, we addressed the problem of learning the structure of Gaussian chain graph models in a high-dimensional space. We argued that when the goal is to recover the model structure, using sparse CGGMs as chain component models has many advantages such as recovery of structured sparsity, computational efficiency, globally-optimal solutions for parameter estimates, and superior performance in semi-supervised learning.

**Acknowledgements**

This material is based upon work supported by an NSF CAREER Award No. MCB-1149885, Sloan Research Fellowship, and Okawa Foundation Research Grant.

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
