[Reviews · NeurIPS 2014]

Submitted by Assigned_Reviewer_11

The paper introduces the case of estimating a sparse chain graphical model in a high-dimensional data setting. It estimates a sparse autoregressive and sparse covariance structure. The method considers one- and multi-levels chain graphical models. The examples and applications are interesting. The authors seem to have missed a recent reference (Abegaz and Wit, Biostatistics, 2013), which exactly deals with sparse chain graphical modelling in multi-level CGMs.

I don't quite understand why the authors did not focus their inference directly on the regression parameters B and partial covariance parameters Theta, as in Abegaz (2013). This would have simplified some inference.
Summary: Chain Gaussian graphical models are convenient ways to describe conditional independence relationships between sets of variables. The paper suggests a sparse approach to CGGM. It would be worth if the authors compared their method to an almost identical method in Abegaz and Wit (2013).

Submitted by Assigned_Reviewer_42

This paper explores how sparse conditional gaussian graphical models (sparse CGGMs) can be used as components of chain graphs. The authors discuss the qualitative and quantitative advantages of using CGGM formulation compared to using sparse multivariate regression with covariance estimation (MRCE) for two and three layer Gaussian chain graph. They go further to show how the framework can easily be extended to handle missing data in the middle layer as semi supervised learning. On their simulation data they show clear advantage of making use of the using CGGM for both network structure reconstruction and for prediction accuracy.

Pros:
The paper is clearly written. It shows how sparse CGGMs can be embedded in a multi layer chain graph structure and covers both structure recovery and inference in the resulting models. The models have intuitive interpretation in the context of genomic data.
The improvement in both structure learning and prediction compared to using linear regression components are clear in the synthetic datasets considered.

Cons:

In general, the authors build heavily on previous works that developed sparse CGGMs (Shon & Kim 2012, Wytock & Kolter 2013). The review style of writing makes it hard to pin down directly from the text what was previously done and what is novel. The authors should state clearly the contributions.

The results/evaluation for real life genomic data evaluation appear limited, cryptic and unconvincing. This makes the method overall less convincing especially as a way to tackle the bio-medical analysis problem they pose. (a) How are the dimensions/features selected? The numbers (e.g. 1000 SNPS, 200 expression "traits"?) seem unrealistic/irrelevant for the general task they offer to tackle originally (millions of SNPS, thousands of genes etc.) (b) How is performance evaluated and what is the meaning of the prediction error values in Table 1? How is "structure" being scored in this setting?? (c) There are no error bars for these results (Table 1) (d) Contrary to the claim in the main text (Line 422) the improvement over LR models appears to be minute to non existing.

The authors comment about the ability to deduce dependency between input features (p. 5 top) but this ability is not explored in their experimental evaluations.

Other comments:

Line 063: "more appropriate" than what?
Line 74-75: "We show that our approach can reveal the module structure ...in different types of genomic data effectively" - this is actually shown only for synthetic data experiments. The relation to real genomic data is not clear.
Line 158: "recently it has been argued" Where? please give a citation.
Line 204: Many of the multi task learning works do not assume the different "tasks" or outputs share the same values from the input features, which is the underlying assumption in the CGGM model. The authors should clarify that point to put their work in the right context.
Figure 2: It seems x6 is irrelevant for the points been made in both Fig2a and 2b. If so consider removing it.
Line 261: "It is well-known that...." is rather a strong statement that implies the relations described always hold. Consider revising (e.g."in many cases...") and adding an appropriate reference(s) that clearly establish the soundness of the claim.

Summary: The authors develop single and three layer chain graphs with sparse CGGMs as their components, discuss how these can be used to infer network structure and apply the method to a combination of genetic (SNP) genomic (gene expression), and phenotype data (pancreatic islet cell).

Submitted by Assigned_Reviewer_43

The paper describes a method to learn sparse chain graph models with Gaussian distributions. It shows simulated and real results that suggest the model works better than using a non-chain graph alternative model.

As I understand the simulation results, the chain-graph model is superior to the GGM when the data comes from a chain-graph model, and the chain-graph model is about the same in terms of P/R as the GGM when the data comes from a GGM. I don't understand why the prediction accuracy is so much better for the chain-graph model when the data comes from a GMM.

It's hard to interpret the mouse-data results. What does a 0.9 prediction error mean? Is this the average number of errors out of the 100 test samples? Is the difference between (e.g.) 1.0661 and 1.0505 important for this domain? Is it statistically significant?
Summary: The paper describes a new way to learn chain-graph models and presents results.
Author Feedback
Author rebuttal: We thank all the reviewers for the valuable comments.

Reviewer 11:

Thank you for bringing to our attention the recent work of Abegaz and Wit (2013). The method by Abegaz and Wit (2013) is different from our method because they build their chain graph model using sparse linear regression components, while we use sparse CGGM components. The chain component model in Abegaz and Wit’s model is identical to the model in MRCE (Rothman et al., 2010) that we discuss in our paper. Abegaz and Wit incorrectly state that the resulting optimization problem with L1 regularization is convex, but it is actually only bi-convex, as proven for the identical approach for MRCE (Yin and Li, 2011). In contrast, our approach leads to convex optimization problem and the optimization algorithms converge more quickly, as we discuss in the paper. We will add a discussion on the comparison of our method with Abegaz and Wit’s, if our paper is accepted.

Reviewer 42:

Regarding the reviewer's comment on the contributions of the paper, we would like to clarify that our contribution lies in the use of sparse CGGM as component models in Gaussian chain graph models. With this construction of Gaussian chain graph models, the model inherits the desirable properties of sparse CGGM such as convexity of the optimization problem and structured output prediction. Furthermore, our model can reveal dependencies among input variables via moralization, which was not possible in sparse CGGMs as they model only a conditional probability ignoring the input structure. Finally, we explore the use of semi-supervised learning with chain graph models for genomic data analysis.

Regarding the reviewer’s comment on the real-data analysis, we were unable to include all the details due to limited space. We plan to include a complete discussion on the real-data analysis in a future journal version of the paper. Below are our responses to the reviewer’s questions and comments.
(a) In response to the reviewer's question about our choice of features, we limited our analysis to several groups of genes with correlated expressions found using hierarchical clustering and SNPs with high variance. We agree that the number of features used was unrealistic. However, this was unfortunately necessary because the optimization method for the alternative model based on linear regression converges very slowly (more than 24 hours), compared to the learning procedure for our model (few minutes).
(b) With regards to the meaning of "prediction error" in Table 1, this is the mean-squared error on the held-out test set. We could not quantitatively assess structure recovery because the ground truth was not available for mouse data, but our method was significantly better at structure recovery on simulated data.
(c) The reviewer requests an assessment of statistical significance on the mouse data, but it is not clear how to do so without performing some kind of subsampling, because only one single dataset is available. However, we had statistically significant results on simulation experiments where we generated 30 independent datasets.
(d) As pointed out by the reviewer, it is not obvious how to assess the significance of the improvement in prediction accuracy on the mouse data. However, our method has additional benefits in dealing with real-world data, such as faster optimization and better sparse support recovery as documented with synthetic data.

In response to the reviewer's request for experimental evaluation of our method's ability to deduce dependency between input features, we qualitatively evaluate performance on this task with simulated data (see Figure 3). We agree that a quantitative evaluation for this would be useful and we will consider including it in our revision.

We will incorporate the reviewer's other comments when updating our paper.

Reviewer 43:

The reviewer incorrectly states that our comparisons are between a Gaussian chain graph model and GGM. Instead, we compare two different methods for sparse Gaussian chain graph models: using sparse linear regression components (previous approaches) and using CGGM components (our approach). Nevertheless, comparison between Gaussian chain graph models and GGMs would be interesting future work.

Regarding our method's superior prediction accuracy even if the true model uses sparse linear regression components, we believe this is largely due to our method's ability to yield a global optimum rather than a local optimum using sparse linear regression. For prediction tasks, the benefit from finding globally optimal parameters significantly outweighs the cost of enforcing sparsity on a different set of parameters than the true model parameters. We will include this discussion in the paper if it is accepted.

Regarding the details on mouse-data results, please see our response to Reviewer 42.